# Deep Functional Dictionaries: Learning Consistent Semantic Structures on 3D Models from Functions

**Minhyuk Sung**
Stanford University
mhsung@cs.stanford.edu

**Hao Su**
University of California San Diego
haosu@eng.ucsd.edu

**Ronald Yu**
University of California San Diego
ronaldyu@ucsd.edu

**Leonidas Guibas**
Stanford University
guibas@cs.stanford.edu

## Abstract

Various 3D semantic attributes such as segmentation masks, geometric features, keypoints, and materials can be encoded as per-point probe functions on 3D geometries. Given a collection of related 3D shapes, we consider how to jointly analyze such probe functions over different shapes, and how to discover common latent structures using a neural network — even in the absence of any correspondence information. Our network is trained on point cloud representations of shape geometry and associated semantic functions on that point cloud. These functions express a shared semantic understanding of the shapes but are not coordinated in any way. For example, in a segmentation task, the functions can be indicator functions of arbitrary sets of shape parts, with the particular combination involved not known to the network. Our network is able to produce a small dictionary of basis functions for each shape, a dictionary whose span includes the semantic functions provided for that shape. Even though our shapes have independent discretizations and no functional correspondences are provided, the network is able to generate latent bases, in a consistent order, that reflect the shared semantic structure among the shapes. We demonstrate the effectiveness of our technique in various segmentation and keypoint selection applications.

## 1   Introduction

Understanding 3D shape semantics from a large collection of 3D geometries has been a popular research direction over the past few years in both the graphics and vision communities. Many applications such as autonomous driving, robotics, and bio-structure analysis depend on the ability to analyze 3D shape collections and the information associated with them.

**Background**   It is common practice to encode 3D shape information such as segmentation masks, geometric features, keypoints, reflectance, materials, etc. as *per-point functions defined on the shape surface*, known as *probe functions*. We are interested, in a joint analysis setting, in discovering common latent structures among such probe functions defined on a collection of related 3D shapes. With the emergence of large 3D shape databases [7], a variety of data-driven approaches, such as cycle-consistency-based optimization [17] and spectral convolutional neural networks [6], have been applied to a range of tasks including semi-supervised part co-segmentation [16, 17] and supervised keypoint/region correspondence estimation [41].

However, one major obstacle in joint analysis is that each 3D shape has its own individual functional space, and linking related functions across shapes is challenging. To clarify this point, we contrast

3D shape analysis with 2D image processing. Under the functional point of view, each 2D image is a function defined on the regular 2D lattice, so all images are functions over a common underlying parameterizing domain. In contrast, with discretized 3D shapes, the probe functions are generally defined on heterogeneous shape graphs/meshes, whose nodes are points on each individual shape and edges link adjacent points. Therefore, the functional spaces on different 3D shapes are independent and not naturally aligned, making joint analysis over the probe functions non-trivial.

To cope with this problem, in the classical framework, ideas from manifold harmonics and linear algebra have been introduced. To analyze meaningful functions that are often smooth, a compact set of basis functions are computed by the eigen-decomposition of the shape graph/mesh Laplacian matrix. Then, to relate basis functions across shapes, additional tools such as functional maps must be introduced [29] to handle the conversions among functional bases. This, however, raises further challenges since functional map estimation can be challenging for non-isometric shapes, and errors are often introduced in this step. In fact, functional maps are computed from *corresponding* sets of probe functions on the two shapes, something which we neither assume nor need.

**Approach**   Instead of a two-stage procedure to first build independent functional spaces and then relate them through correspondences (functional or traditional), we propose a novel correspondence-free framework that directly learns consistent bases across a shape collection that reflect the shared structure of the set of probe functions. We produce a compact encoding for meaningful functions over a collection of related 3D shapes by learning a small functional basis for each shape using neural networks. The set of functional bases of each shape, a.k.a *a shape-dependent dictionary*, is computed as a set of functions on a point cloud representing the underlying geometry — a functional set whose span will include probe functions on that shape. The training is accomplished in a very simple manner by giving the network sequences of pairs consisting of a shape geometry (as point clouds) and a semantic probe function on that geometry (that should be in the associated basis span). Our shapes are correlated, and thus the semantic functions we train on reflect the consistent structure of the shapes. The neural network will maximize its representational capacity by learning consistent bases that reflect this shared functional structure, leading in turn to consistent sparse function encodings. Thus, in our setting, consistent functional bases *emerge* from the network without explicit supervision.

We also demonstrate how to impose different constraints to the network optimization problem so that atoms in the dictionary exhibit desired properties adaptive to application scenarios. For instance, we can encourage the atoms to indicate *smallest* parts in the segmentation, or *single* points in keypoint detection. This implies that our model can serve as a collaborative filter that takes any mixture of semantic functions as inputs, and find the finest granularity that is the shared latent structure. Such a possibility can particularly be useful when the annotations in the training data are *incomplete and corrupted*. For examples, users may desire to decompose shapes into specific parts, but all shapes in the training data have only partial decomposition data without labels on parts. Our model can aggregate the partial information across the shapes and learn the full decomposition.

We remark that our network can be viewed as a function autoencoder, where the decoding is required to be in a particular format (a basis selection in which our function is compactly expressible). The resulting *canonicalization* of the basis (the consistency we have described above) is something also recently seen in other autoencoders, for example in the quotient-space autoencoder of [10] that generates shape geometry into a canonical pose.

In experiments, we test our model with existing neural network architectures, and demonstrate the performance on labeled/unlabeled segmentation and keypoint correspondence problem in various datasets. In addition, we show how our framework can be utilized in learning synchronized basis functions with random continuous functions.

**Contribution**   Though simple, our model has advantages over the previous bases synchronization works [37, 36, 41] in several aspects. First, our model does not require precomputed basis functions. Typical bases such as Laplacian (on graphs) or Laplace-Beltrami (on mesh surfaces) eigenfunctions need extra preprocessing time for computation. Also, small perturbation or corruption in the shapes can lead to big differences. We can avoid the overhead of such preprocesssing by *predicting* dictionaries while also synchronizing them simultaneously. Second, our dictionaries are application-driven, so each atom of the dictionary itself can attain a semantic meaning associated with small-scale geometry, such as a small part or a keypoint, while LB eigenfunctions are only suitable for approximating continuous and smooth functions (due to basis truncation). Third, the previous works

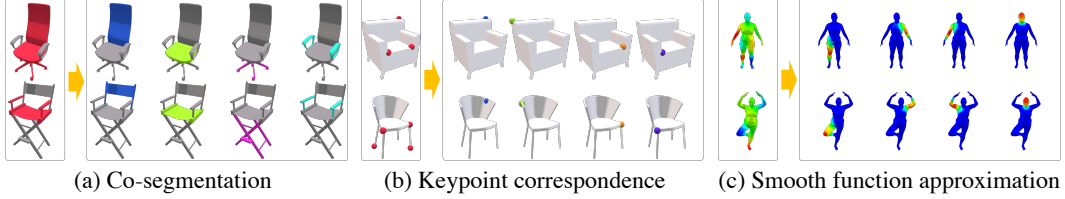

| (a) Co-segmentation | (b) Keypoint correspondence | (c) Smooth function approximation |

Figure 1: Inputs and outputs of various applications introduced in Section 3: (a) co-segmentation, (b) keypoint correspondence, and (c) smooth function approximation problems. The inputs of (a) and (b) are a random set of segments/keypoints (without any *labels*), and the outputs are single segment/keypoint per atom in the dictionaries consistent across the shapes. The input of (c) is a random linear combination of LB bases, and the outputs are synchronized atomic functions.

define *canonical* bases, and the synchronization is achieved from the mapping between each individual set of bases and the canonical bases. In our model, the neural network becomes the synchronizer, without any explicit canonical bases. Lastly, compared with classical dictionary learning works that assume a universal dictionary for all data instances, we obtain a *data-dependent dictionary* that allows non-linear distortion of atoms but still preserves consistency. This endows us additional modeling power without sacrificing model interpretability.

## 1.1 Related Work

Since much has already been discussed above, we only cover important missing ones here.

Learning compact representations of signals has been widely studied in many forms such as factor analysis and sparse dictionaries. Sparse dictionary methods learn an overcomplete basis of a collection of data that is as succinct as possible and have been studied in natural language processing [9, 12], time-frequency analysis [8, 22], video [25, 1], and images [21, 42, 5]. Encoding sparse and succinct representations of signals has also been observed in biological neurons [27, 26, 28].

Since the introduction of functional maps [29], shape analysis on functional spaces has also been further developed in a variety of settings [30, 20, 17, 11, 34, 24], and mappings between pre-computed functional spaces have been studied in a deep learning context as well [23]. In addition to our work, deep learning on point clouds has also been done on shape classification [32, 33, 19, 39], semantic scene segmentation [15], instance segmentation [38], and 3D amodal object detection [31]. We bridge these areas of research in a novel framework that learns, in a data-driven end-to-end manner, data-adaptive dictionaries on the functional space of 3D shapes.

## 2 Problem Statement

Given a collection of shapes $\{\mathcal{X}_i\}$, each of which has a sample function of specific semantic meaning $\{f_i\}$ (e.g. indicator of a subset of semantic parts or keypoints), we consider the problem of sharing the semantic information across the shapes, and predicting a functional dictionary $A(\mathcal{X};\Theta)$ for each shape that linearly spans all plausible semantic functions on the shape ($\Theta$ denotes the neural network weights). We assume that a shape is given as $n$ points sampled on its surface, a function $f$ is represented with a vector in $\mathbb{R}^n$ (a scalar per point), and the atoms of the dictionary are represented as columns of a matrix $A(\mathcal{X};\Theta) \in \mathbb{R}^{n \times k}$, where $k$ is a sufficiently large number for the size of the dictionary. Note that the column space of $A(\mathcal{X};\Theta)$ can include any function $f$ if it has all Dirac delta functions of all points as columns. We aim at finding a much lower-dimensional vector space that also contains all plausible semantic functions. We also force the columns of $A(\mathcal{X};\Theta)$ to encode *atomic* semantics in applications, such as atomic instances in segmentation, by adding appropriate constraints.

## 3 Deep Functional Dictionary Learning Framework

**General Framework** We propose a simple yet effective loss function, which can be applied to any neural network architecture processing a 3D geometry as inputs. The neural network takes pairs of a shape $\mathcal{X}$ including $n$ points and a function $f \in \mathbb{R}^n$ as inputs in training, and outputs a matrix $A(\mathcal{X};\Theta) \in R^{n \times k}$ as a dictionary of functions on the shape. The loss function needs to be designed

1: **function** SINGLE STEP GRADIENT ITERATION($\mathcal{X}, f, \Theta^t, \eta$)
2:    Compute: $A^t = A(\mathcal{X}; \Theta^t)$.
3:    Solve: $\boldsymbol{x}^t = \arg\min_{\boldsymbol{x}} \|A^t \boldsymbol{x} - f\|_2^2 \quad \text{s.t.} \quad C(\boldsymbol{x})$.
4:    Update: $\Theta^{t+1} = \Theta^t - \eta \nabla L(A(\mathcal{X}; \Theta^t); f, \boldsymbol{x}^t)$.
5: **end function**

**Algorithm 1:** Single-Step Gradient Iteration. $\mathcal{X}$ is an input shape ($n$ points), $f$ is an input function defined on $\mathcal{X}$, $\Theta^t$ is neural network weights at time $t$, $A(\mathcal{X}; \Theta^t)$ is an output dictionary of functions on $\mathcal{X}$, $C(\boldsymbol{x})$ is the constraints on $\boldsymbol{x}$, and $\eta$ is learning rate. See Section 2 and 3 for details.

for minimizing both 1) the projection error from the input function $f$ to the vector space $A(\mathcal{X}; \Theta)$, and 2) the number of atoms in the dictionary matrix. This gives us the following loss function:

$$L(A(\mathcal{X}; \Theta); f) = \min_{\boldsymbol{x}} F(A(\mathcal{X}; \Theta), \boldsymbol{x}; f) + \gamma \|A(\mathcal{X}; \Theta)\|_{2,1}$$

$$\text{s.t.} \quad F(A(\mathcal{X}; \Theta), \boldsymbol{x}; f) = \|A(\mathcal{X}; \Theta)\boldsymbol{x} - f\|_2^2 \qquad (1)$$

$$C(A(\mathcal{X}; \Theta), \boldsymbol{x}),$$

where $\boldsymbol{x} \in \mathbb{R}^k$ is a linear combination weight vector, $\gamma$ is a weight for regularization. $F(A(\mathcal{X}; \Theta))$ is a function that measures the projection error, and the $l_{2,1}$-norm is a regularizer inducing *structured* sparsity, encouraging more columns to be zero vectors. We may have a set of constraints $C(A(\mathcal{X}; \Theta), \boldsymbol{x})$ on both $A(\mathcal{X}; \Theta)$ and $\boldsymbol{x}$ depending on the applications. For example, when the input function is an *indicator* (binary) function, we constrain all elements in both $A(\mathcal{X}; \Theta)$ and $\boldsymbol{x}$ to be in $[0, 1]$ range. Other constraints for specific applications are also introduced at the end of this section.

Note that our loss minimization is a min-min optimization problem; the inner minimization, which is embedded in our loss function in Equation 1, optimizes the reconstruction coefficients based on the shape dependent dictionary predicted by the network, and the outer minimization, which minimizes our loss function, updates the neural network weights to predict a best shape dependent dictionary. The nested minimization generally does not have an analytic solution due to the constraint on $\boldsymbol{x}$. Thus, it is not possible to directly compute the gradient of $L(A(\mathcal{X}; \Theta); f)$ without $\boldsymbol{x}$. We solve this by an alternating minimization scheme as described in Algorithm 1. In a single gradient descent step, we first minimize $F(A(\mathcal{X}; \Theta); f)$ over $\boldsymbol{x}$ with the current $A(\mathcal{X}; \Theta)$, and then compute the gradient of $L(A(\mathcal{X}; \Theta); f)$ while fixing $\boldsymbol{x}$. The minimization $F(A(\mathcal{X}; \Theta); f)$ over $\boldsymbol{x}$ is a convex quadratic programming, and the scale is very small since $A(\mathcal{X}; \Theta)$ is a very thin matrix ($n \gg k$). Hence, a simplex method can very quickly solve the problem in every gradient iteration.

**Adaptation in Weakly-supervised Co-segmentation**    Some constraints for both $A(\mathcal{X}; \Theta)$ and $x$ can be induced from the assumptions of the input function $f$ and the properties of the dictionary atoms. In the segmentation problem, we take an indicator function of a set of segments as an input, and we desire that each atom in the output dictionary indicates an atomic part (Figure 1 (a)). Thus, we restrict both $A(\mathcal{X}; \Theta)$ and $x$ to have values in the $[0, 1]$ range. Also, the atomic parts in the dictionary must *partition* the shape, meaning that each point must be assigned to one and only one atom. Thus, we add sum-to-one constraint for every *row* of $A(\mathcal{X}; \Theta)$. The set of constraints for the segmentation problem is defined as follows:

$$C_{\text{seg}}(A(\mathcal{X}; \Theta), \boldsymbol{x}) = \begin{cases} \mathbb{0} \leq \boldsymbol{x} \leq \mathbb{1} \\ \mathbb{0} \leq A(\mathcal{X}; \Theta) \leq \mathbb{1} \\ \sum_j A(\mathcal{X}; \Theta)_{i,j} = 1 \text{ for all } i \end{cases}, \qquad (2)$$

where $A(\mathcal{X}; \Theta)_{i,j}$ is the $(i, j)$-th element of matrix $A(\mathcal{X}; \Theta)$, and $\mathbb{0}$ and $\mathbb{1}$ are vectors/matrices with an appropriate size. The first constraint on $\boldsymbol{x}$ is incorporated in solving the inner minimization problem, and the second and third constraints on $A(\mathcal{X}; \Theta)$ can simply be implemented by using softmax activation at the last layer of the network.

**Adaptation in Weakly-supervised Keypoint Correspondence Estimation**    Similarly with the segmentation problem, the input function in the keypoint correspondence problem is also an indicator function of a set of points (Figure 1 (b)). Thus, we use the same $[0, 1]$ range constraint for both $A(\mathcal{X}; \Theta)$ and $x$. Also, each atom needs to represent a *single* point, and thus we add sum-to-one constraint for every *column* of $A(\mathcal{X}; \Theta)$:

$$C_{\text{key}}(A(\mathcal{X};\Theta), \boldsymbol{x}) = \begin{cases} \mathbb{0} \leq \boldsymbol{x} \leq \mathbb{1} \\ \mathbb{0} \leq A(\mathcal{X};\Theta) \leq \mathbb{1} \\ \sum_i A(\mathcal{X};\Theta)_{i,j} = 1 \text{ for all } j \end{cases} \qquad (3)$$

For robustness, a distance function from the keypoints can be used as input instead of the binary indicator function. Particularly some neural network architectures such as PointNet [32] do not exploit local geometry context. Hence, a spatially localized distance function can avoid overfitting to the Dirac delta function. We use a normalized Gaussian-weighed distance function $g$ in our experiment: $g_i(s) = \frac{\exp(d(p_i,s)^2/\sigma)}{\sum_i g_i(s)}$, where $g_i(s)$ is $i$-th element of the distance function from the keypoint $s$, $p_i$ is $i$-th point coordinates, $d(\cdot,\cdot)$ is Euclidean distance, and $\sigma$ is Gaussian-weighting parameter (0.001 in our experiment). The distance function is normalized to sum to one, which is consistent with our constraints in Equation 3. The sum of any subset of the keypoint distance functions becomes an input function in our training.

**Adaptation in Smooth Function Approximation and Mapping**    For predicting atomic functions whose linear combination can approximate any smooth function, we generate the input function by taking a random linear combination of LB bases functions (Figure 1 (c)). We also use a unit vector constraint for each atom of the dictionary:

$$C_{\text{map}}(A(\mathcal{X};\Theta), \boldsymbol{x}) = \left\{ \sum_i A(\mathcal{X};\Theta)_{i,j}^2 = 1 \text{ for all } j \right\} \qquad (4)$$

# 4    Experiments

We demonstrate the performance of our model in keypoint correspondence and segmentation problems with different datasets. We also provide qualitative results of synchronizing atomic functions on non-rigid shapes. While any neural network architecture processing 3D geometry can be employed in our model (e.g. PointNet [32], PointNet++ [33], KD-NET [19], DGCNN [39], ShapePFCN [18]), we use PointNet [32] architecture in the experiments due to its simplicity. Note that our output $A(\mathcal{X};\Theta)$ is a set of $k$-dimensional row vectors for all points. Thus, we can use the PointNet segmentation architecture without any modification. Code for all experiments below is available in https://github.com/mhsung/deep-functional-dictionaries.

## 4.1    ShapeNet Keypoint Correspondence

Yi et al. [41] provide keypoint annotations on 6,243 chair models in ShapeNet [7]. The keypoints are manually annotated by experts, and all of them are matched and aligned across the shapes. Each shape has up to 10 keypoints, while most of the shapes have missing keypoints. In the training, we take a random subset of keypoints of each shape to feed an input function, and predict a function dictionary in which atoms indicate every single keypoint. In the experiment, we use a 80-20 random split for training/test sets [1], train the network with $2k$ point clouds as provided by [41], and set $k = 10$ and $\gamma = 0.0$.

Figure 2 (at the top) illustrates examples of predicted keypoints when picking the points having the maximum value in each atom. The colors denote the order of atoms in dictionaries, which is consistent across all shapes despite their different geometries. The outputs are also evaluated by the percentage of correct keypoints (PCK) metric as done in [41] while varying the Euclidean distance threshold (Figure 2 at the bottom). We report the results for both when finding the best one-to-one correspondences between the

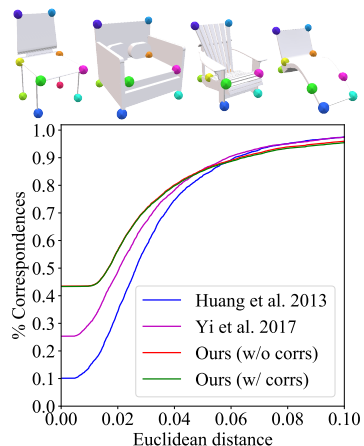

Figure 2: ShapeNet keypoint correspondence result visualizations and PCK curves.

Table 1: ShapeNet part segmentation comparison with PointNet segmentation (same backbone network architecture as ours). Note that PointNet has additional supervision (class labels) compared with ours (Sec 4.2). The average mean IoU of our method is measured by finding the correspondences between ground truth and predicted segments for each shape. $k = 10$ and $\gamma = 1.0$.

| | mean | air-plane | bag | cap | car | chair | ear-phone | guitar | knife | lamp | laptop | motor-bike | mug | pistol | rocket | skate-board | table |
|---|---|---|---|---|---|---|---|---|---|---|---|---|---|---|---|---|---|
| PointNet [32] | 82.4 | **81.4** | **81.1** | 59.0 | 75.6 | 87.6 | 69.7 | **90.3** | **83.9** | 74.6 | 94.2 | **65.5** | **93.2** | 79.3 | 53.2 | 74.5 | 81.3 |
| Ours | **84.6** | 81.2 | 72.7 | **79.9** | 76.5 | 88.3 | 70.4 | 90.0 | 80.5 | **76.1** | 95.1 | 60.5 | 89.8 | **80.8** | **57.1** | 78.3 | 88.1 |

Table 2: ShapeNet part segmentation results. The first row is when finding the correspondences between ground truth and predicted segments per shape. The second row is when finding the correspondences between part labels and indices of atoms per category. $k = 10$ and $\gamma = 1.0$.

| | mean | air-plane | bag | cap | car | chair | ear-phone | guitar | knife | lamp | laptop | motor-bike | mug | pistol | rocket | skate-board | table |
|---|---|---|---|---|---|---|---|---|---|---|---|---|---|---|---|---|---|
| Ours (per shape) | **84.6** | **81.2** | **72.7** | **79.9** | **76.5** | **88.3** | **70.4** | 90.0 | **80.5** | **76.1** | **95.1** | **60.5** | **89.8** | **80.8** | **57.1** | **78.3** | **88.1** |
| Ours (per cat.) | 77.3 | 79.0 | 67.5 | 66.9 | 75.4 | 87.8 | 58.7 | **90.0** | 79.7 | 37.1 | 95.0 | 57.1 | 88.8 | 78.4 | 46.0 | 75.8 | 78.4 |

ground truth and predicted keypoints for each shape (red line)
and when finding the correspondences between ground truth labels and atom indices for all shapes (green line). These two plots are identical, meaning that the order of predicted keypoints is rarely changed in different shapes. Our results also outperform the previous works [14, 41] by a big margin.

## 4.2 ShapeNet Semantic Part Segmentation

ShapeNet [7] contains 16,881 shapes in 16 categories, and each shape has semantic part annotations [40] for up to six segments. Qi et al. [32] train PointNet segmentation using shapes in all categories, and the loss function is defined as the cross entropy per point with all labels. We follow their experiment setup by using the same split of training/validation/test sets and the same $2k$ sampled point cloud as inputs. The difference is that we do not leverage the *labels* of segments in training, and consider the parts as *unlabeled* segments. We also deal with the more general situation that each shape may have *incomplete* segmentation by taking an indicator function of a random subset of segments as an input.

**Evaluation**   For evaluation, we binarize $A(\mathcal{X}; \Theta)$ by finding the maximum value in each row, and consider each column as an indicator of a segment. The accuracy is measured based on the average of each shape mean IoU similarly with Qi et al. [32], but we make a difference since our method does not exploit *labels*. In ShapeNet, some categories have *optional* labels, and shapes may or may not have a part with these optional labels (e.g. armrests of chairs). Qi et al. [32] take into account the optional labels even when the segment does not exist in a shape [2]. But we do not predict labels of points, and thus such cases are ignored in our evaluation.

We first measure the performance of segmentation by finding the correspondences between ground truth and predicted segments for *each shape*. The best one-to-one correspondences are found by running the Hungarian algorithm on mean IoU values. Table 1 shows the results of our method when using $k = 10$ and $\gamma = 1.0$, and the results of the label-based PointNet segmentation [32]. When only considering the segmentation accuracy, our approach outperforms the original PointNet segmentation trained with labels.

We also report the average mean IoUs when finding the best correspondences between part labels and the indices of dictionary atoms *per category*. As shown in Table 2, the accuracy is still comparable in most categories, indicating that the order of column vectors in $A(\mathcal{X}; \Theta)$ are mostly consistent with the semantic labels. There are a few exceptions; for example, lamps are composed of shades, base, and tube, and half of lamps are ceiling lamps while the others are standing lamps. Since PointNet learns per-point features from the global coordinates of the points, shades and bases are easily confused when their locations are switched (Figure 3). Such problem can be resolved when using a different

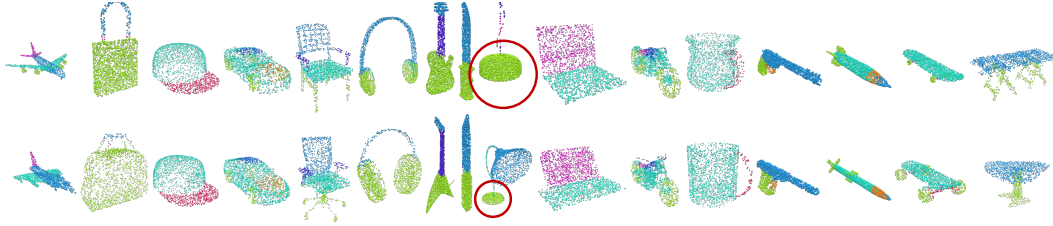

Figure 3: Examples of ShapeNet part segmentation results. The colors indicate the indices of atoms in the dictionaries. The order of atoms are consistent in most shapes except when the part geometries are not distinguishable. See the confusion of a ceiling lamp shade (at first row) and a standing lamp base (at second row) highlighted with red circles.

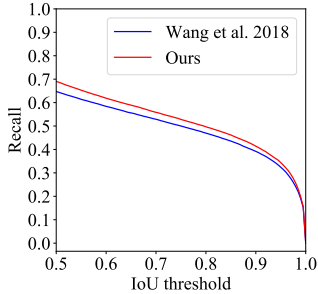

Figure 4: S3DIS instance segmentation proposal recall comparison while varying IoU threshold.

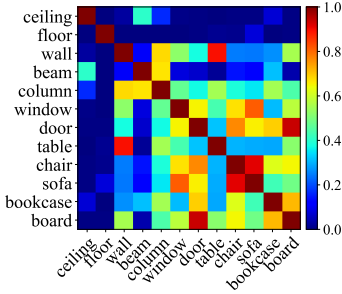

Figure 5: S3DIS instance segmentation confusion matrix for ground truth object labels.

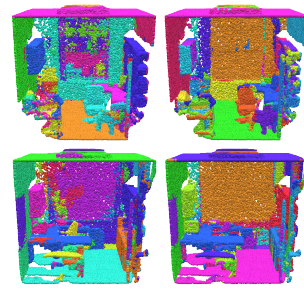

Figure 6: Comparison of S3DIS instance segmentation results. Left is SGPN [38], and right is ours.

Table 3: S3DIS instance segmentation proposal recall comparison per class. IoU threshold is 0.5.

| | mean | ceiling | floor | wall | beam | column | window | door | table | chair | sofa | bookcase | board |
|---|---|---|---|---|---|---|---|---|---|---|---|---|---|
| SGPN [38] | 64.7 | 67.0 | 71.4 | 66.8 | **54.5** | **45.4** | 51.2 | **69.9** | **63.1** | **67.6** | **64.0** | **54.4** | **60.5** |
| Ours | **69.1** | **95.4** | **99.2** | **77.3** | 48.0 | 39.2 | **68.2** | 49.2 | 56.0 | 53.2 | 35.3 | 31.6 | 42.2 |

neural network architecture learning more from the local geometric contexts. For more analytic experiments, refer to the supplementary material.

## 4.3 S3DIS Instance Segmentation

Stanford 3D Indoor Semantic Dataset (S3DIS) [2] is a collection of real scan data of indoor scenes with annotations of instance segments and their semantic labels. When segmenting instances in such data, the main difference with the semantic segmentation of ShapeNet is that there can exist multiple instances of the same semantic label. Thus, the approach of classifying points with labels is not applicable. Recently, Wang et al. [38] tried to solve this problem by leveraging the PointNet architecture. Their framework named SGPN learns a similarity metric among points, enabling every point to generate a instance proposal based on proximity in the learned feature space. The per-point proposals are further merged in a heuristic post processing step. We compare the performance of our method with the same experiment setup with SGPN. The input is a $4k$ point cloud of a $1m \times 1m$ floor block in the scenes, and each block contains up to 150 instances. Thus, we use $k = 150$ and $\gamma = 1.0$. Refer to [38] for the details of the data preparation. In the experiments of both methods, all 6 areas of scenes except area 5 are used as a training set, and the area 5 is used as a test set.

**Evaluation** We evaluate the performance of instance proposal prediction in each block of the scenes. [3] As an evaluation metric, we use proposal recall [13], which measures the percentage

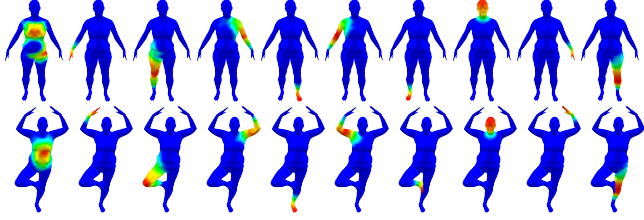
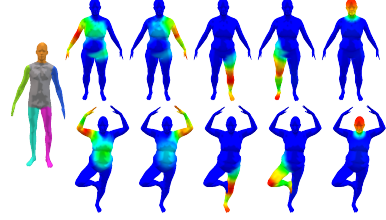

Figure 7: Output atomic functions with random continuous functions on MPI-FAUST human shapes [4]. $K = 10$ and $\gamma = 0.0$. The order of atoms are consistent.

Figure 8: Five parts transfer from the base shape (left) to other shapes (each row).

of ground truth instances covered by any prediction within a given IoU threshold. In both SGPN and our model, the outputs are non-overlapped segments, thus we do not consider the number of proposals in the evaluation. Figure 4 depicts the proposal recall of both methods when varying the IoU threshold from 0.5 to 1.0. The recall of our method is greater than the baseline throughout all threshold levels. The recalls for each semantic part label with IoU threshold 0.5 are reported in Table 3. Our method performs well specifically for large objects such as ceilings, floors, walls, and windows. Note that Wang et al. [38] start their training from a pretrained model for semantic label prediction, and also their framework consumes point labels as supervision in the training to jointly predict labels and segments. Our model is trained from scratch and label-free.

**Consistency with semantic labels**    Although it is hard to expect strong correlations among semantic part labels and the indices of dictionary atoms in this experiment due to the large variation of scene data, we still observe weak consistency between them. Figure 5 illustrates confusion among semantic part labels. This confusion is calculated by first creating a vector for each label in which the $i$-th element indicates the count of the label in the $i$-th atom, normalizing this vector, and taking a dot product for every pair of labels. Ceilings and floors are clearly distinguished from the others due to their unique positions and scales. Some groups of objects having similar heights (e.g. doors, bookcases, and boards; chairs and sofas) are confused with each other frequently, but objects in different groups are discriminated well.

## 4.4   MPI-FAUST Human Shape Bases Synchronization

In this experiment, we aim at finding synchronized atomic functions in a collection of shapes for which linear combination can approximate any continuous function. Such synchronized atomic functions can be utilized in transferring any information on one shape to the other without having point-wise correspondences. Here, we test with 100 non-rigid human body shapes in MPI-FAUST dataset [4]. Since the shapes are deformable, it is not appropriate to process Euclidean coordinates of a point cloud as inputs. Hence, instead of a point cloud and the PointNet, we use HKS [35] and WKS [3] point descriptors for every vertex, and process them using 7 residual layers shared for all points as proposed in [23]. The point descriptors cannot clearly distinguish symmetric parts in a shape, so the output atomic functions also become symmetric. To break the ambiguity, we sample four points using farthest point sampling in each shape, find their one-to-one correspondences in other shapes using the same point descriptor, and use the geodesic distances from these points as additional point features. As input functions, we compute Laplace-Beltrami operators on shapes, and take a random linear combination of the first ten bases.

Figure 7 visualizes the output atomic function when we train the network with $k = 10$ and $\gamma = 0.0$. The order of atomic functions are consistent in all shapes. In Figure 8, we show how the information in one shape is transferred to the other shapes using our atomic functions. We project the indicator function of each segment (at left in figure) to the function dictionary space of the base shape, and unproject them in the function dictionary space of the other shapes. The transferred segment functions are blurry since the network is trained with only continuous functions, but still indicate proper areas of the segments.

## 5   Conclusion

We have investigated a problem of jointly analyzing probe functions defined on different shapes, and finding a common latent space through a neural network. The learning framework we proposed predicts a function dictionary of each shape that spans input semantic functions, and finds the atomic functions in a consistent order without any correspondence information. Our framework is very general, enabling easy adaption to any neural network architecture and any application scenario. We have shown some examples of constraints in the loss function that can allow the atomic functions to have desired properties in specific applications: the smallest parts in segmentation, and single points in keypoint correspondence.

In the future, we will further explore the potential of our framework to be applied to various applications and even in different data domains. Also, we will investigate how the power of neural network decomposing a function space to atoms can be enhanced through different architectures and a hierarchical basis structure.

**Acknowledgments**

We thank the anonymous reviewers for their comments and suggestions. This project was supported by a DoD Vannevar Bush Faculty Fellowship, NSF grants CHS-1528025 and IIS-1763268, and an Amazon AWS AI Research gift.

## Footnotes

[1]Yi et al. [41] use a select subset of models in their experiment, but this subset is not provided by the authors. Thus, we use the entire dataset and make our own train/test split.

[2]IoU becomes zero if the label is assigned to any point in prediction, and one otherwise.

[3]Wang et al. [38] propose a heuristic process of merging prediction results of each block and generating instance proposals in a scene, but we measure the performance for *each block* in order to factor out the effect of this post-processing step.

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
