[Supplementary Material]

# Deep Functional Dictionaries: Learning Consistent Semantic Structures on 3D Models from Functions – Supplementary Material

**Minhyuk Sung**
Stanford University
mhsung@cs.stanford.edu

**Hao Su**
University of California San Diego
haosu@eng.ucsd.edu

**Ronald Yu**
University of California San Diego
ronaldyu@ucsd.edu

**Leonidas Guibas**
Stanford University
guibas@cs.stanford.edu

## S.1  ShapeNet Semantic Part Segmentation – Analytic Experiments

**Effect of $k$ and $\gamma$**  In Table S1, we demonstrate the effect of changing parameter $k$ and $\gamma$. When the $l_{2,1}$-norm regularizer is not used ($\gamma = 0$), the accuracy decreases as $k$ increases since parts can map to a number of smaller segments. After adding the regularizer with a weight $\gamma$, the accuracy becomes similar however we choose the number of columns $k$. We found that the $l_{2,1}$-norm regularizer effectively forces the unnecessary columns to be close to a zero vector.

**Training with partial segmentation**  In the segmentation problem using *unlabeled* segments, learning from partial segmentation is a non-trivial task, while our method can easily learn segmentation from the partial information. To demonstrate this, we randomly select a set of parts in the entire training set with a fixed fraction, and ignore them when choosing a random subset of segments for input functions. The accuracy according to the fraction is shown in Table S2. Note that performance remains roughly the same even when we do not use 75% of segments in the training.

**Training with noise**  We test the robustness of our training system against noise in the input function $b$. Table S3 describes the performance when switching each bit of the binary indicator function with a specific probability. The results show that our system is not affected by the small noise in the input functions.

Table S1: Average mIoU on ShapeNet parts with different $k$ and $\gamma$

| $k$ \ $\gamma$ | 0.0 | 0.5 | 1.0 |
|---|---|---|---|
| 10 | **75.0** | 82.7 | 84.6 |
| 25 | 71.2 | **83.8** | **85.2** |
| 50 | 65.3 | 82.9 | 82.9 |

Table S2: Average mIoU on ShapeNet parts with partial segmentations ($k = 10, \gamma = 1.0$).

| Fraction | mIoU |
|---|---|
| 0.00 | 84.6 |
| 0.25 | **86.1** |
| 0.50 | 86.0 |
| 0.75 | 84.5 |

Table S3: Average mIoU on ShapeNet parts with noise in inputs ($k = 10, \gamma = 1.0$).

| Probability | mIoU |
|---|---|
| 0.00 | 84.6 |
| 0.05 | 85.8 |
| 0.10 | **85.9** |
| 0.20 | 85.1 |

## S.2  Siamese Structure for Correspondence Supervision

While our framework empirically performs well on generating consistent function dictionaries even without correspondences, we further investigate about how the correspondence supervision can be

incorporated in our framework when it is provided. We consider the case when the correspondence information is given as a pair or functions in different shapes. Note that this setup does not require to have full correspondence information for *all* pairs. The correspondence of functions means that two functions are represented with the same linear combination weight $x$ when the order of dictionary atoms are consistent. Thus, we build a Siamese neural network structure processing two corresponding functions, and minimize the inner problem $F(A(\mathcal{X};\Theta),x;f)$ in the loss function Equation 1 jointly with the shared variable $x$.

We test this approach with the ShapeNet part segmentation problem. Every time when feeding the input function in the training, we find the other shape that have a corresponding function, and randomly choose one of them. The comparison with the vanilla framework is shown in Table S4 and S5. $k = 10$ and $\gamma = 1.0$ are used in both experiments. When finding the best one-to-one correspondences between ground truth part labels and atom indices in each *category*, the Siamese structure shows 3.0% improvement in average mean IoU, meaning that the output dictionaries make less confusion when distinguishing semantic parts with the indices of atoms. It also gives better accuracy when finding the correspondences in each *shape*.

Table S4: Performance comparison of vanilla and Siamese structures when finding the correspondences between part labels and atom indices per *category*. $k = 10$ and $\gamma = 1.0$.

| | mean | air-plane | bag | cap | car | chair | ear-phone | guitar | knife | lamp | laptop | motor-bike | mug | pistol | rocket | skate-board | table |
|---|---|---|---|---|---|---|---|---|---|---|---|---|---|---|---|---|---|
| Vanilla | 77.3 | **79.0** | 67.5 | **66.9** | 75.4 | **87.8** | 58.7 | 90.0 | 79.7 | 37.1 | **95.0** | 57.1 | 88.8 | 78.4 | 46.0 | 75.8 | 78.4 |
| Siamese | **80.3** | 78.6 | **73.7** | 44.8 | **76.9** | 87.7 | **65.0** | **90.6** | **85.2** | **60.4** | 94.7 | **60.5** | **93.6** | **78.5** | **55.8** | **76.1** | **80.1** |

Table S5: Performance comparison of vanilla and Siamese structures when finding the correspondences between part labels and atom indices per *object*. $k = 10$ and $\gamma = 1.0$.

| | mean | air-plane | bag | cap | car | chair | ear-phone | guitar | knife | lamp | laptop | motor-bike | mug | pistol | rocket | skate-board | table |
|---|---|---|---|---|---|---|---|---|---|---|---|---|---|---|---|---|---|
| Vanilla | 84.6 | 81.2 | 72.7 | **79.9** | 76.5 | 88.3 | 70.4 | 90.0 | 80.5 | 76.1 | 95.1 | 60.5 | 89.8 | **80.8** | 57.1 | 78.3 | 88.1 |
| Siamese | **85.6** | **82.2** | **75.7** | 74.5 | **77.5** | **88.4** | **73.5** | **91.0** | **85.2** | **77.9** | **95.9** | **63.4** | **93.6** | 80.7 | **62.4** | **80.7** | **88.9** |