[Reviews · NeurIPS 2018]

Reviewer 1



The authors propose a loss function for training a neural network such that it can jointly analyze 3D shapes and the associated probe functions in a correspondence free manner. The neural network takes as input pairs of shapes, represented as point clouds, and functions and learns a dictionary for each shape that linearly spans all plausible semantic functions on the shape. The devised loss function penalizes the projection error from the input function to learned space, while at the same time imposing structured sparsity on the learned dictionary. The proposed approach was tested in three tasks: key point correspondence, part segmentation, and smooth function approximation. This was possible by appropriately adapting the imposed constraints in the objective function. In the tested tasks, the proposed approach performed favorably when compared to baseline approaches. The paper is well-organized and written. The proposed technique is interesting and well motivated, albeit it is not fully clear how to implement it. The proposed methodology is fairly evaluated across a number of different tasks. However, the comparison with state of the art methods is scarce. Supplementary results suggest that the method is robust and can also take into consideration provided correspondences to improve the performance.

Reviewer 2



Main idea --------- In this work, the authors use a neural network to discover common latent structures in a given collection of related 3D shapes. They formalize the problem as finding shape-dependent dictionaries of basis functions that are meant to capture semantic information shared among all shapes. Interestingly, the aim is to discover such structures even in the absence of any correspondence information. Strengths - Contributions ------------------------- Introduction is well written and related work seems well described. 1. The model does not require (costly) precomputed basis functions (such as Laplace-Beltrami eigenfunctions) as most approaches. 2. Dictionaries are application-driven, atoms can approximate non-continuous / non-smooth functions. 3. No need to explicitly define canonical bases to synchronize shapes, the neural network becomes the synchronizer. 4. No assumption of a universal dictionary, a data-dependent dictionary allows non-linear distortion of atoms but still preserves consistency. The results seem to improve upon recent works. Weaknesses ---------- The main section of the paper (section 3) seems carelessly written. Some obvious weaknesses: - Algorithm 1 seems more confusing, than clarifying: a) Shouldn't the gradient step be taken in the direction of the gradient of the loss with respect to Theta? b) There is no description of the variables, most importantly X and f. It is better for the reader to define them in the algorithm than later in the text. Otherwise, the algorithm definition seems unnecessary. - Equation (1) is very unclear: a) Is the purpose to define a loss function or the optimization problem? It seems that it is mixing both. b) The optimization variable x is defined to be in R^n. Probably it is meant to be in R^k? c) The constraints notation (s.t. C(A, x)) is rather unusual. - It is briefly mentioned that an alternating direction method is used to solve the min-min problem. Which method? - The constraints in equation (2) are identical to the ones in equation (3). They can be mentioned as such to gain space. - In section 4.1, line 194, K = 10, presumably refers to the number of atoms in the dictionary, namely it should be a small k? The same holds for section 4.4, line 285. - In section 4.1, why is the regularizer coefficient gamma set to zero? Intuitively, structured sparcity should be particularly helpful in finding keypoint correspondences. What is the effect on the solution when gamma is larger than zero? The experimental section of the paper seems well written (with a few exceptions, see above). Nevertheless, the experiments in 4.2 and 4.3 each compare to only one existing work. In general, I can see the idea of the paper has merit, but the carelessness in the formulations in the main part and the lack of comparisons to other works make me hesitant to accept it as is at NIPS.

Reviewer 3



The main contribution of this work is a deep neural network approach to learning the shared structure of probe functions (functions defined on each point of a 3D surface) for a shape collection. The result is a dictionary of basis functions that spans all probe functions on that shape. A loss function for learning (with a gradient computation technique) is also presented for this approach, and particular applications to segmentation, key point correspondence estimation, and shape synchronization. Experiments are performed in several datasets and domains with favorable results on average. === Strengths === 1) The idea of learning this type of general encoding with NN is interesting and, as far as I know, novel. The paper does a good job in showing the wide applicability of the approach by providing specific constraint functions for different problems, as well as by showing experimental results in each domain. 2) The experiments are complete and informative. Several datasets and domains were explored, and comparison against state-of-the-art methods was presented. The results themselves also show significant improvements. All of this supports the premise of the paper, and shows strong evidence that the approach is effective. === Weaknesses === 1) The impact of this work is very limited. Although the model seems to be quite general, it is not clear how difficult it is to implement, nor if it will consistently provide advantages over approaches specifically designed for each domain. Perhaps more experiments would help make this more clear. 2) Presentation. Although the approach is clearly explained, and technical details are roughly at the appropriate level, the paper lacks a discussion as to why this method works well. That is, what is it about these learned bases that make it effective? It would be nice to have some kind of hypothesis here that could be supported by the experimental results. === Questions === 1) Section 4.3 (roughly L267-270) briefly explains why some objects are confused with others (doors and bookcases, etc.). Can this be addressed in this approach? Why does SPGN seem to do better in several of these object categories (Table 3)? 2) Why are there no quantitative results for the Human Shape Bases Synchronization problem? The qualitative results look impressive, but it is difficult to judge effectiveness on only a few pictures. === Conclusion === The paper presents an interesting, novel approach to learning bases/dictionaries for probe functions, and provides sufficient experimental evidence to show effectiveness. The only substantial negative is the limited impact of the approach. I think the paper is borderline, but I lean towards acceptance.